# Neural network-based Bluetooth synchronization of multiple wearable devices

Karthikeyan Kalyanasundaram Balasubramanian [1] ✉, Andrea Merello[1], Giorgio Zini[1], Nathan Charles Foster[2], Andrea Cavallo [2,3], Cristina Becchio [2,4] & Marco Crepaldi [1] ✉

Bluetooth-enabled wearables can be linked to form synchronized networks to provide insightful and representative data that is exceptionally beneficial in healthcare applications. However, synchronization can be affected by inevitable variations in the component's performance from their ideal behavior. Here, we report an application-level solution that embeds a Neural network to analyze and overcome these variations. The neural network examines the timing at each wearable node, recognizes time shifts, and fine-tunes a virtual clock to make them operate in unison and thus achieve synchronization. We demonstrate the integration of multiple Kinematics Detectors to provide synchronized motion capture at a high frequency (200 Hz) that could be used for performing spatial and temporal interpolation in movement assessments. The technique presented in this work is general and independent from the physical layer used, and it can be potentially applied to any wireless communication protocol.

The adoption of Bluetooth-based wearable devices for the collection of physiological data in real-world scenarios[1,2] has seen a steady increase, primarily due to their portability, convenience, and safety attributes. Most applications, including those within the health sector[3], require collection from multiple devices. However, current time synchronization methods for Bluetooth Low Energy (BLE) multi-channel systems cannot satisfy the requirement of accurate mutual synchronization (i.e., multiple wearables performing a group activity in unison). Here, we present an application-level solution for Bluetooth synchronization to remotely control wearable devices and ensure mutual synchronization without placing an undue burden on the hardware.

Bottlenecks from the BLE protocol are mostly due to channel jamming and multiple-user functionality. Integrating these wearable devices with an ad hoc protocol (i.e., ANT+, WiFi, Zigbee) in the 2.4 GHz industrial, scientific, and medical (ISM) band of the wireless medium can solve the problem. However, if a wearable includes an ad hoc wireless protocol (thus replacing BLE), power consumption figures and physical size of the wearable device normally increase, and a non-standardized auxiliary receiver at the remote end is required because these wireless protocols are typically proprietary. Given the widespread use of BLE in many electronic devices (e.g., laptops, mobiles, and tablets), wireless synchronization across multiple wearables via BLE is preferable because it maintains standardized and secured connectivity. BLE-based wearable device synchronization is achievable with specific hardware-software co-design, as detailed in this work.

BLE is an asynchronous wireless protocol for data transfer and operates assuming either an advertising or a connected mode between two electronic entities. In advertisement mode, a one-to-many network transmission occurs based on the Generic Access Profile (GAP)[4], particularly on a broadcast protocol with no data coherence, security, or encryption. Data on the advertisement header is accessible to every device on the network, thus raising concerns for data security[5]. As the General Data Protection Regulation (GDPR) becomes mandatory when

[1]Electronic Design Laboratory (EDL), Istituto Italiano di Tecnologia, Genova, Italy. [2]Cognition, Motion and Neuroscience (C'MON), Istituto Italiano di Tecnologia, Genova, Italy. [3]Department of Psychology, University of Turin, Torino, Italy. [4]Department of Neurology, University Medical Centre Hamburg-Eppendorf, Hamburg, Germany. ✉e-mail: karthikeyan.kalyanasundaram@iit.it; marco.crepaldi@iit.it

handling personal health data and biometric information, wearables must adhere to the data security regulations, especially while capturing data outside hospital environments or laboratory settings. On the other hand, when BLE operates in connected mode, the data is encrypted before wireless transmission, thus complying with the Generic Attribute Transfer (GATT) protocol from the BLE specifications[4]. The electronics in this mode can accept and acknowledge recurring messages from the remote system to carry out an assigned task (e.g., ascertain predefined or user-defined characteristics to implement a given functionality). For instance, reading the battery status from a wearable using GATT can be achieved using predefined characteristics $0 \times 2A19$ through a return value in the range of 0–100%. GATT protocol is a suitable option for wearables[6]. It enables additional data security and customization while sending commands (a feature that can be enabled or programmed), resulting in increased flexibility in controlling wearable functionality. Real-Time Operating Systems (RTOSs) help improve the flexibility in developing wearable devices. Active multitasking and pre-emptive scheduling, advanced power supply control, high responsiveness with predictability, enhanced protection, and a streamlined code development and maintenance approach make RTOSs excellent candidates for standalone wearable systems. To operate with RTOSs, each component in the hardware has to be conceptually classified, systematically outlined, and prioritized for execution. Unlike general-purpose operating systems, the task scheduler in the RTOSs is tightly designed on top of the hardware for appropriate device functionality; thereby, it relies heavily on the system clock and hardware time parameters for performing any designed tasks.

Time-sensitive parameters at these wearables represent direct factors that impact multiple device synchronization. Temperature, power, process variations, aging, and crystal reference drifts can degrade their optimal performance, influencing their timing accuracy and contributing to unwanted delays (e.g., propagation delay, device latency, and device clock[7,8]). Enhancing time references in a wearable is a viable solution for improving multiple device synchronization, even using high-level protocols. In this respect, Clock Synchronization Protocols (CSPs) such as Flood Time Synchronization Protocol (FTSP)[9], Reference Broadcast Synchronization (RBS)[10], and Timing Sync Protocols for Sensor Networks (TPSN)[11] can be effectively used to achieve synchronization. Moreover, FTSP features have become a de-facto standard in many CSPs, such as using time stamps (i.e., storing recording time for every communication event across the associated devices). Notably, CSPs forecast three remedial strategies to improve synchronization, i.e., (1) fine-tuning wearable clocks before the experiment, (2) differing clock ticks at runtime[12], and (3) applying a time delay[13].

However, when dealing with energy-constrained portable systems, applying high-level protocols directly to microcontrollers to alter low-level parameters, such as directly modifying their hardware clock, would inevitably fail in the presence of RTOSs. Indeed, both multitasking mechanisms and scheduling operate based on time-slicing concepts. When the device time or clock is altered, the device would eventually miss critical interrupts or send out-of-sync signals to the hardware sub-systems, thus leading to hard failures. Last but not least, besides hardware-related time-sensitive parameters, the intrinsic non-determinism of the Bluetooth stack, encompassing anchor points (i.e., the events at which Bluetooth transmission wakes up for asynchronous data transfer) and retries, lead to an unpredictable time shift during packet transmission, resulting in non-uniform device operation thus affecting mutual synchronization among multiple instances.

To address these limitations and improve the quality of synchronization, we designed a neural network (NN, inside the remote end to control wearable devices using BLE) consisting of weights and biases that identify a pattern from a time-stamped series, including various sources of time shifts and non-deterministic parameters, from both wearable non-ideal behavior and wireless communication channel. This NN enables the prediction of various sources of time shifts from physical components, operating system scheduling-related issues, and wireless mediums considered as a whole.

## Results

We aim to demonstrate multiple Bluetooth-based wearable device synchronization in a setup featuring Kinematics Detectors[1] (KiD, see Fig. 1a) as hardware and using them across a developed User Interface (UI) as software. KiD is a high-performance motion-tracking device specifically designed for real-world applications. Its lightweight, small form factor and design make it suitable for capturing limb movement in both toddlers[14] and adults, applicable to multi-limb and multi-agent settings. KiD can continuously acquire data for two and a half hours (150 min), and it is specifically designed for short-duration in-lab assessments (see Supplementary Note, "Wearables design methodology"). Furthermore, KiD supports the BLE 4.2 protocol (see Supplementary Note, "KiD Hardware layout" for the hardware details) and has reconfigurable firmware, allowing users to reconfigure parameters in the hardware subroutines wherever necessary to interact with the software. Our hardware-software co-design goal here is to provide a universal solution for synchronizing Bluetooth-based devices without having an additional burden on the hardware, particularly for

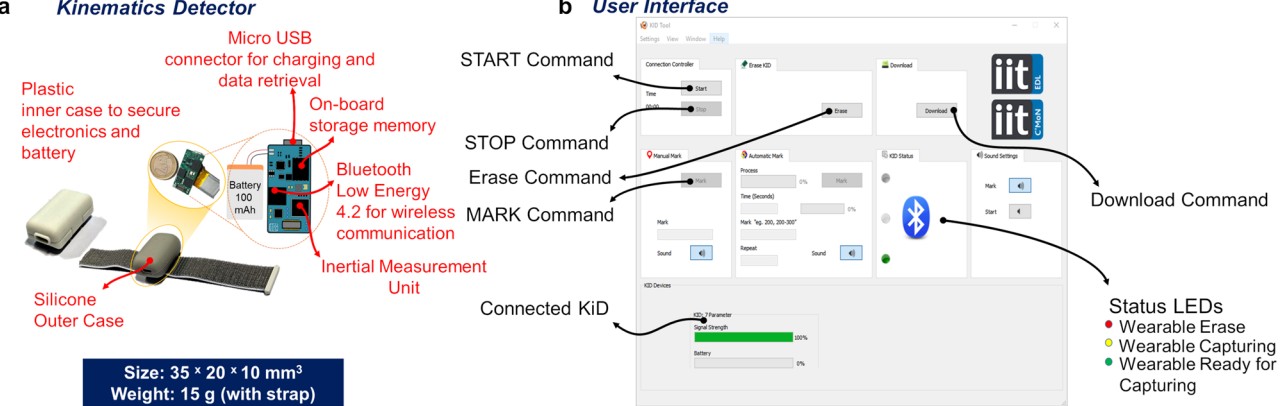

**Fig. 1 | Kinematics Detector, a motion capture device and its User Interface.** **a** KiD motherboard with electronics and battery, wherein they are encapsulated using a plastic enclosure, and silicon outer case to be worn on the wrist using a hook and fastening strap. **b** User Interface that controls the wearables for experimenting. START begins the experiment, and STOP suspends the experiment. The UI provides means to know the wearable's operation status and information like the physical distance from the remote system and information on the wearable's battery level, which is crucial while performing experiments.

synchronization. The designed software sends commands to a synchronous network of wearables to enable easy control and maintenance.

The evaluator can remotely control the devices by sending commands from the remote system using the UI (see Fig. 1b). Here, the participants can wear the KiD directly on their limbs for an experiment and perform a prescribed task given by the evaluator. The UI is designed to display the wearable's current status, perform any maintenance task (clearing the inbuilt memory or renaming the wearable if

required), control them for a certain experiment, download their data (in a comma separated value format), and eventually report communication and internal errors to troubleshoot any potential issues (see Supplementary Methods, "KiD Software flow chart"). Figure 2a depicts the motion capture operational scheme with one or more KiDs. The depicted scheme gives an example of designing an experiment using these wearables. For straightforward data capture, the commands between the remote system and the devices are sent over BLE 4.2. Bluetooth communication between devices is based on START and

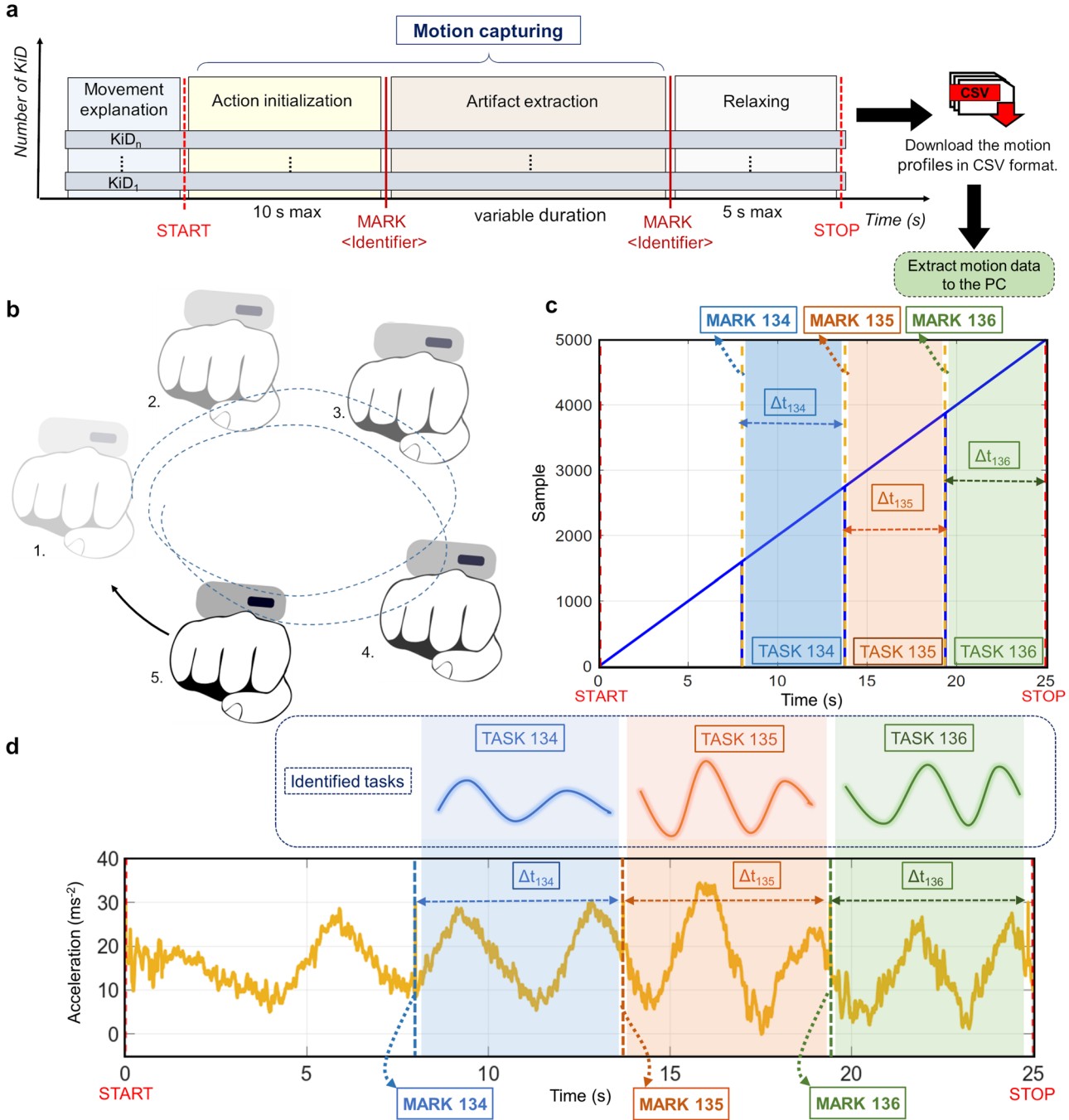

**Fig. 2 | Motion capturing that facilitates online labels to be sent as task identifiers during acquisition. a** `Motion capturing` operational scheme that demonstrates the use of KiDs regardless of the number of devices linked to the User Interface. **b** An example hand motion sequence used in the mocap demonstration. **c** Timeline and the number of samples captured using KiD for an artifact for 25 s, where the frequency of the KiD is $199.846 \pm 0.1430$ Hz and R-value = 0.9987 when

the test is repeated 100 times. **d** Examples of IMU data (motion profile) that were captured within START to STOP events from the limbs. The colored window represents the tagged sections of TASK 134, TASK 135, TASK 136 that were sent during the motion capture using the MARK feature. The time for the task is denoted as $\Delta t_{134} = 7$ s, $\Delta t_{135} = 5.2$ s, and $\Delta t_{136} = 5.8$ s.

`STOP` commands (see Supplementary Table 1) that perform `Action initialization` and suspend data capture. Let us consider that the evaluator wants to perform an investigation to capture a hand motion sequence (rotating the hand clockwise for 25 s, see Fig. 2b). In this case, the evaluator fixes a KiD onto a participant's wrist, performs configuration, instructs the participant about the execution of the desired motion sequence, and through the UI, the experimenter sends `START` and `STOP` commands as illustrated on the aforementioned operational scheme for `Motion capturing`. Figure 2c, d shows the data profile of such task, downloaded from the KiD using the UI, where the acquired 'sample' or data are captured within the reception of the commands `START` and `STOP` by the wireless device. The `MARK` feature enables the tagging of captured data with labels, where the labels are stored directly onto the KiD's memory wirelessly to identify an event or task while processing the data. When the `Motion capturing` experiment is repeated over 100 times, the sampling frequency of the device has been found to have a mean of $199.846 \pm 0.1430$ Hz, with an R-value = 0.9987, thus confirming high sampling frequency (see "Methods: Low-level parameter evaluation").

## Multiple device synchronization

Quantifying time uncertainty in the devices can provide a better over-view of how profoundly they affect the synchronous operation. In such cases, the devices shall abide to have identical or designated data to derive the time deviations during the multiple device data capture. In particular, each KiD shall have the designated spatial domain across all devices to support measuring time uncertainty with a less complicated or simple interpretation. One feasible technique to achieve the designated spatial domain is to fasten every wearable to a fixed plane[12]. KiDs,

in this case, are stacked vertically (see Fig. 3a, b, where each KiD is placed on top of another) in a fixed plane for acquiring the given rotation. Given that the device's operation is already known (here, KiD collects the temporal and spatial information from an IMU at 200 Hz; see Supplementary Fig. 4, KiD as a standalone system), the `MARK` commands are sent periodically while capturing. At the devices, these `MARK` commands are automatically stored inside the device's memory upon receiving them. For every sent `MARK`, time in the device, the reaction time of the device, time in the remote system, and their differences are systematically noted and organized. For instance, Fig. 3c shows the stacked KiDs captured data, i.e., the acceleration profiles in terms of the spatial and temporal domain are the designated spatial attributes. `MARK 435` is one such command, and its timing is systematically noted across devices for calculating temporal shifts ($\tau$, the time difference between the devices on a common reference frame). Figure 3d depicts the temporal shifts among the devices, where these shifts indicate that the devices transmit data asynchronously while using Bluetooth.

As mentioned previously, propagation delay, Bluetooth re-transmission, anchor points, and frequency drifts are limiting factors in the Bluetooth technology (mostly present while using GATT) and prevent the synchronous operation of the wearables, thus resulting in non-linear and in-determinable temporal shifts. To achieve synchronization, the devices must ideally run timing error-free, have zero propagation delay, have time-independent software routines, and maintain relationships among devices during the data acquisition session.

## Temporal-shift evaluation

The factors behind such temporal shifts must be precisely identified and tackled. First, in a multiple-device synchronous network,

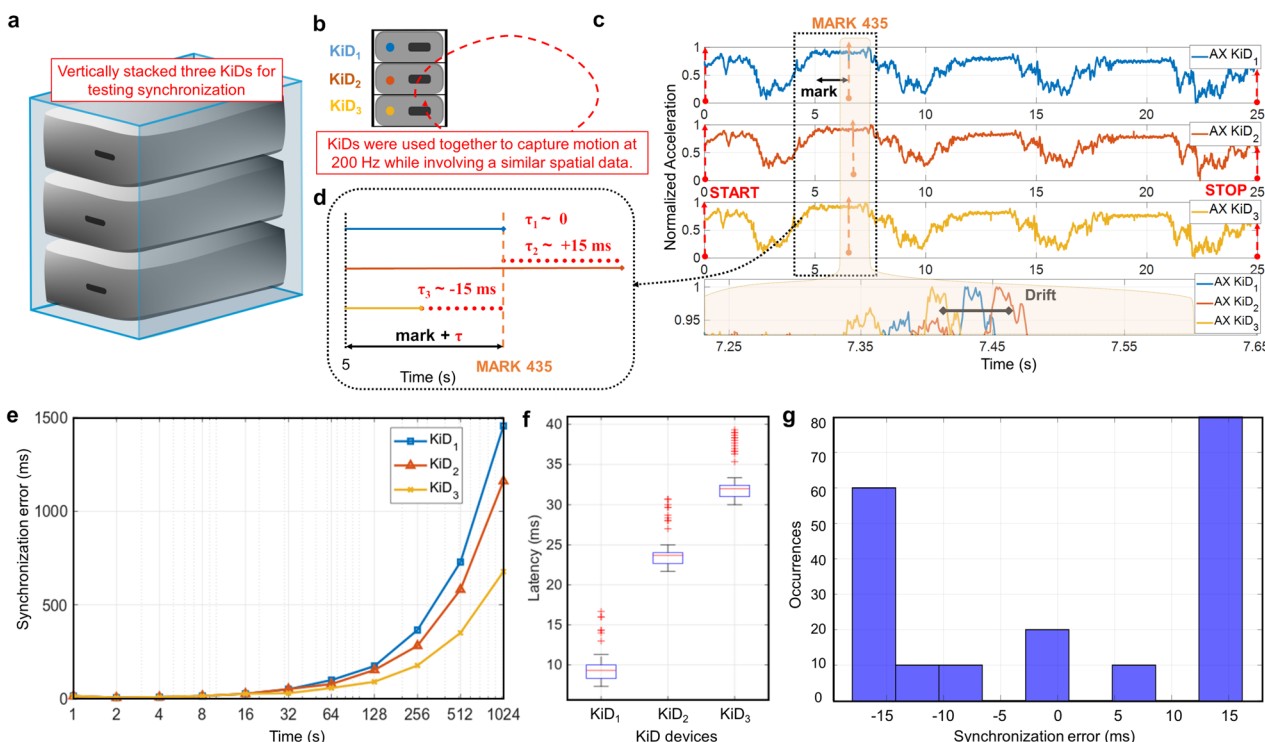

**Fig. 3 | Multiple device evaluation without synchronization algorithm. a** KiDs are vertically stacked upon each other, i.e., $KiD_i$ where $i$ is 1…3 for evaluating the multiple device performance. **b** While data capturing, all these KiDs share the same spatial attributes, allowing it more straightforward to analyze the temporal attribute. **c** Acceleration profiles from three KiDs show the 200 Hz motion data with a similar spatial attribute. **d** The temporal shifts ($\tau_i$ where $i$ is 1…3 indicate temporal shifts in each wearable device) within KiDs provide incorrect time information; thus, they provide inaccurate time stamping. **e** Frequency drift among three

devices $KiD_i$ where $i$ is 1…3 when connected to the remote system without synchronization for 1024 s. **f** Example of KiDs latency, when connected with three devices, due to the Bluetooth transmission for a synchronized message that affects synchronization, the mean values found to be $KiD_1 = 9.5 \pm 1.762$ ms, $KiD_2 = 23.8 \pm 2.01$ ms, $KiD_3 = 32.5 \pm 2.37$ ms. **g** The average synchronization error in these wearables is 30 ms, i.e., the frequency drift within devices and the Bluetooth latency contribute to this indeterministic error causing inaccurate time interpolation while experimenting.

frequency drift among wearables is critical (see "Methods" section on Evaluating communication). For instance, Fig. 3e shows the frequency drift of three devices compared to the system clock. The sampling frequency of $KiD_1$, $KiD_2$, and $KiD_3$ is 199.68 Hz, 199.72 Hz, and 199.82 Hz, respectively. Second, each command requires transmission and acknowledgment by the devices resulting in a given latency. We have sampled the delay required to transmit packets between the remote system and devices to estimate propagation delay, which directly influences latency. As shown in Fig. 3f, the latency is a non-determinable Bluetooth parameter due to the Bluetooth stack re-transmission (retry protocol: see Supplementary Fig. 5, Bluetooth stack operation), which is influenced by the conditions of the wireless medium. The overall impact of the temporal shift is an increased average synchronization error (see Fig. 3g, where the average synchronization error is 30 ms), which leads to unsynchronized wearable devices.

## Neural network for synchronization

Often, the only way to evaluate non-determinable temporal shifts is using a non-linear estimator (e.g., a neural network) that, in this context, can be trained and devised to adjust packet transmission times and hence counterbalance the uncertain delay caused by the wireless transmission. Figure 4a shows how KiDs are attached to the neural network[15]. For instance, consider three devices, $KiD_1$, $KiD_2$, and $KiD_3$, connected to the network. When the host system sends a command to these devices, it does it sequentially from one device to another at a given system time ($t_i^M$). Each device accepts the command at different device times ($t_i^S$) (see Evaluating wireless communication channel in the "Methods" section for the relationship between devices and `Remote System` and for details about virtual clock concepts).

Figure 4b depicts the neural network output (see Supplementary Methods, "Neural network concepts" and Supplementary Fig. 6) while learning the time relationship between the remote system and the

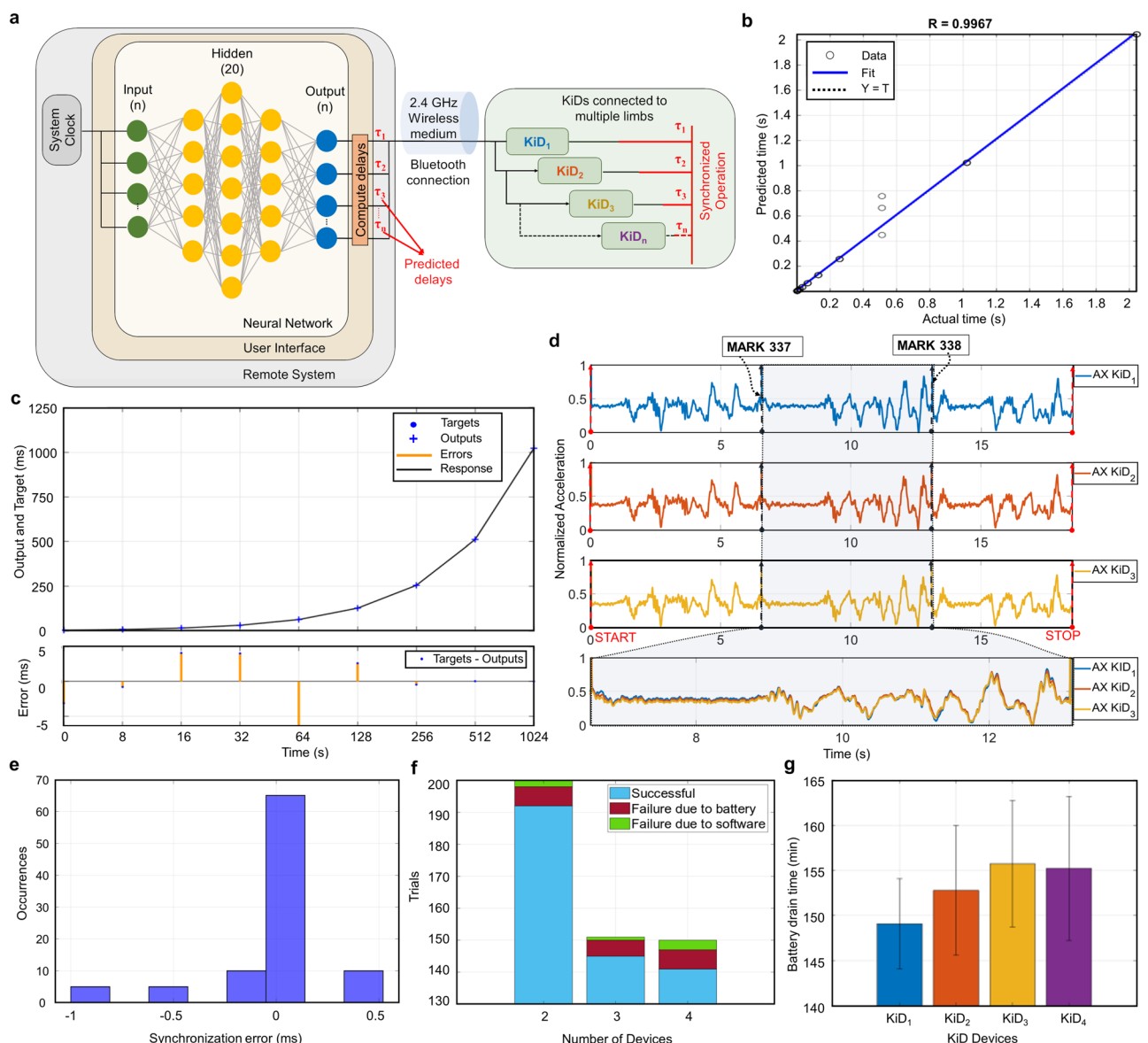

**Fig. 4 | Neural network interpretation and implementation. a** Neural network implementation technique at the application level for improving synchronization. **b** Correlation between the predicted and actual time where the R-value is 0.9967. **c** Neural network time response and error graph. **d** Three perfectly synchronized KiDs for interpolating temporal and spatial attributes contributing toward detecting fine motion skills between MARK 337 and MARK 338. The accelerometer (raw data of one dimension) is normalized to explain synchronization. **e** Improved average synchronization error to 1.25 ms. **f** Synchronization validation across multiple devices in several platforms (PC, Linux, and Mac) show 97.3% success in synchronization. **g** Battery drain time while synchronized motion capturing.

devices (i.e., the neural network acts as a virtual clock layer) to provide accurate device synchronization. The R-value, i.e., the correlation between the actual and predicted values after 100 iterations, is 0.9967 (see Fig. 4b). The `Compute delays` can precisely estimate $\tau_1$, $\tau_2$, and $\tau_3$, used across the devices to store the message into its device memory at a compensated time. As a result, the devices can adjust frequency drift if present, the device latency, and the communication delays to provide mutual synchronization, as shown in Fig. 4c.

Considering the concept of synchronization, which involves the phase alignment of each remote system clock, sending commands to multiple wearables to synchronize events exactly is a challenging task. Instead, we send labeling events to the synchronous network containing multiple wearables (two or more KiDs linked together via neural networks) through our neural network-based layer, which enables consistent operation across wearables, operating as an additive synchronization layer. For instance, if the evaluator sends `START` to the network through the UI, the wearables in the network instantly invoke the given commands at corrected time instants, thus eliminating the need for the evaluator to send multiple devices individually and accounting for latencies and delays. Under such accounts of operation, Fig. 4d shows the synchronized event from three KiD devices, where these wearables operate synchronously with low delay variation. The `MARK` command automatically transmits the marker to all the KiDs. For example, `MARK 337` and `MARK 338` are communicated through the UI to the three KiDs using the neural network layer, which transmits signals in the correct sequence and compensates for latency and delay across all the connected devices. Thus, all the devices linked to the synchronized network can receive messages from the UI in mutual synchronization. The devices captured data synchronously (see the data in the inset from `MARK 337` to `MARK 338`) and thereby show the significance of reduced temporal shifts. The average synchronization error across the devices is 0.356 ms (see Fig. 4e), allowing it to be suitable for synchronous BLE applications. A lower average synchronization error than the sampling frequency signifies the capacity to take messages while capturing data, notwithstanding that lower average synchronization errors could improve the accuracy of a message transmission[5].

The synchronization network was tested across multiple KiDs (i.e., two, three, and four devices) by repeating the tasks in the range of 150–200 iterations (see Fig. 4f). As shown in the results, however, in some cases, synchronization failed during the trials with a 2.7% failure rate, and those failures were due to two primary reasons that we have addressed to improve our system. The first reason is an OS software update, mostly affecting the UI. We have observed that OS libraries strongly affected timing and performance. Consequently, to address this issue, we have rewritten some parts of the UI to use independent software modules from the OS. The second reason is that the experiment was purposefully carried out to evaluate the outcome of complete battery exhaustion (see Fig. 4g, where the KiDs, on average, work better if the experiments are within 150 min). In both situations, synchronization failed due to device acquisition failure. Consequently, we have updated the UI error and status message management. In particular, the timeout functions now report any synchronization error if the commands are not acknowledged within a given time (see Supplementary Fig. 2).

### Artifact extraction and labeling

Wearables can potentially capture sensor data at high frequencies (e.g., in our case, 200 Hz), resulting in massive volumes of data that demand human intervention (data hard to comprehend if the experiment includes various extracted features[16]) to extract a specific task. Such practices physically limit the experiment to a static time and make feature extraction of a particular event difficult for the evaluator. Therefore, the capacity to easily identify and name the data segments during real-time acquisition can be of significant added value. Thus, we

have added a feature (thanks to the capability of GATT for accepting messages that can edit the datapath of the microcontroller for performing an assigned task) that consists of sending virtual labels during data capture, which we herein define as 'Labeling'. Here, the evaluator can assign a unique label for each task and send it as a message from the remote system to the wearables that store the label alongside the associated data. Data within two unique labels can be easily associated with a given task, providing advantages at higher-level data interpretations, classifications, and clustering. In this context, we have implemented a specific `MARK` command with a label implemented using a 16-bit unsigned arithmetic integer (0-32767) that can be sent to the KiDs through the UI. For instance, `MARK 134` in the motion profile (see Supplementary Table 2), as depicted in Fig. 2c, is one such label sent via the UI, where the label appends to the device's internal memory. KiD(s) continuously acquire data until they receive the `STOP` command. With the help of this low-level primitive, the evaluator can use as many labels as required, thus providing complete annotations to identify critical events. For example, Fig. 2d depicts the intervals between two such labels, that is, $\Delta t_{134}$, i.e., the delay between markers `MARK 134` and `MARK 135`, which contains 7 s of 1400 'samples'/motion data representing a specific task `TASK 134`. Similarly, another interval $\Delta t_{135} = 5.2$ s between `MARK 135` and `MARK 136` identifies `TASK 135` and $\Delta t_{136} = 5.8$ s identifies `TASK 136`.

Labeling (i.e., Event or Artifact extraction) provides a straightforward approach for marking intervals and deciphering factual information from complete unlabeled data[16]. Figure 5 demonstrates a multiple KiD motion capture task comprising four devices ($KiD_1$, $KiD_2$, $KiD_3$, and $KiD_4$), all linked to the NN layer and stacked vertically, demonstrating scalability while using NN. For Artifact extraction in multiple KiDs, the evaluator can use the same command `MARK` followed by an arithmetic label sent to the synchronized network instead of a device. `Identified tasks` in Fig. 5 show the systematic acquisition of `TASK 401`, `TASK 402`, `TASK 403`, and `TASK 404` using the `MARK` command sent to the network. Our 'synchronized labels' (the labels sent to the synchronized network) for identifying tasks among the motion data seem a convenient and crucial method for extracting or performing task-specific group activities (sometimes recognizing critical movements or tasks). The artifacts from multiple limb motion data may be efficiently bundled for any sophisticated numerical models, such as machine learning or artificial intelligence[17], or saved as a significant data set of an anticipated activity. Our labeling method has been conceived to be elementary to allow the investigators to focus on task organization (event coordination or planning) rather than concentrating on the technical aspects of the wearable. We believe our method could be a one-of-a-kind method in data acquisition.

## Discussion

Well-designed, transferable time synchronization methods are needed to support Bluetooth-based wearable devices. Here, we present an application-level solution for Bluetooth-based device synchronization in a setup featuring multiple unsynchronized KiDs. Measuring and quantifying complex, naturalistic, unrestrained, and minimally shaped behavior presents formidable conceptual and technical challenges, which are only just beginning to be addressed in behavioral and neuroscience research[14,18–21] (see Supplementary Discussion, "Motor behavior or sensorimotor analysis"). To tackle this challenge, it is often crucial to collect synchronized responses from multiple devices. KiDs operate solely based on the GATT mode, receiving device commands to perform specific services and securely storing sensor data in their inbuilt memory (not providing an external medium, see Supplementary Discussion, "Safety measures on wearables design for toddlers"). Our synchronized network concept, combined with a method for sending multiple labels, provides a solution for collecting temporal-sensitive responses and synchronizing KiD devices. This capability opens up new avenues for research, ranging from investigations of

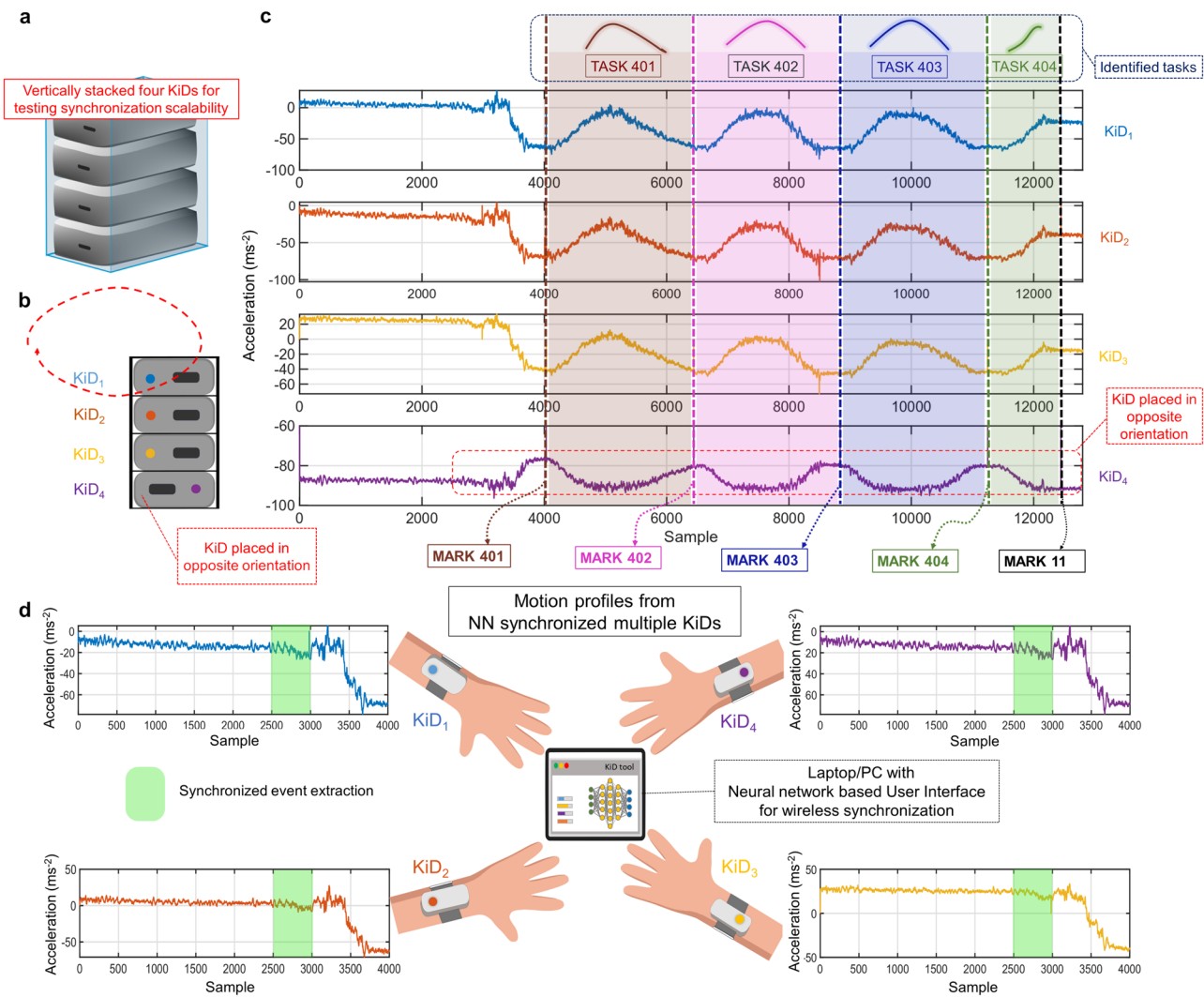

**Fig. 5 | Multiple device synchronization using KiDs while operating via the User Interface with neural network. a** A schematic of four KiDs vertically stacked to illustrate scalability, and **b** the results show the four KiDs synchronization, where the artifact or task extraction reveals the level of synchronization from the KiDs using the trained neural network. MARK 401, MARK 402, MARK 403, MARK 404, and MARK 11 are the online synchronized labels that were sent via UI to identify corresponding tasks demonstrating the synchronization of multiple devices in a motion-tracking study. These commands were sent to the synchronized network connected with four KiDs. **c** For illustrative purposes in the spatial and temporal, we have stacked the four devices, where the fourth device was kept in the opposite direction. **d** The multiple limb motion capture can be used for capturing motions, particularly in real-world settings. KiD, in this method, provides synchronized motion profiles with the online task or event extraction, where in this case, they acquired a motion profile for 20 s throughout the entire duration.

bilateral motor coordination to interpersonal synchronization during real-life multi-agent interactions[22].

Few studies report on wireless sensor or BLE synchronization, and most of them require low-level parameters handling or configuring individual wearables with different settings and do not provide a universal solution for the scope of our development. Altering low-level parameters in the hardware can provide sub-millisecond accuracy. However, it could thwart the entire wearable operation, particularly while using RTOS for the following points. (1) Timer-critical subroutines: waiting a certain amount of time before performing a task or triggering an event would fail as the device change could affect operation timing, causing them to happen earlier or later than expected. (2) Event synchronization: Events and semaphores/signals used to synchronize tasks and communicate between them fail because device time change may affect them, causing tasks to wait longer or shorter than expected. (3) Multi-task scheduling: multi-tasks work based on time-slicing concepts when the device time or clock is altered, which would eventually miss critical interrupts or send untimely signals. (4) Time-stamped data: in some

devices, data is time-stamped to record when it was created or updated. Changing the device time may affect the accuracy of these time stamps, causing data to appear out of order or with incorrect time stamps.

The connection interval, a fundamental parameter of all Bluetooth stack communications, influences the temporal shift or delay. The connection interval has a minimal value of 7.5 ms, and raising it further increases the wearable's latency (see Supplementary Fig. 1). This interval is a bottleneck in numerous wearables today for synchronous applications, limiting their functionality to a maximum of 120 Hz (see Supplementary Fig. 8, Parameter comparison). Since Bluetooth transmission is digital and packet-based (therefore not pertaining to a continuous analog transmission), it attempts to deliver the command to the devices more than once in the event of a transmission failure. The two factors mentioned above are predominant in causing indeterministic temporal shifts that impact mutual synchronization at devices with an average synchronization error of 30 ms, as depicted in Fig. 3f. This variability is inherently difficult to predict and remains a significant drawback of Bluetooth, particularly when dealing

with multiple devices that need to operate in unison. Moreover, when wearables are interfaced with a general-purpose computer, other Operating System (OS) related scheduling issues can additionally contribute to these time shifts because OS's kernels cannot always be real-time in all applications.

Our work demonstrates the capability of the NN layer to solve synchronization issues at the application level. The proposed solution requires additional Bluetooth-transferred data to acquire clock ticks for time stamp recording during a command transfer. In this context, a wearable must draw additional power for every extra communication the remote system requires. However, our data show that the additional Bluetooth communication load with each wearable KiD device was limited and did not impact battery life, which was unvaried compared to the initial design requirements, i.e., 150 min acquisition session.

In synchronized networks, many Bluetooth-based wearables are connected and work together to obtain event-oriented sensor data. The NN inside the software interface can be configured to be autonomously trained with the device parameters (by automating supervised learning tasks through the UI) and thus sending the commands on time for synchronization without any additional human effort, providing intrinsic scalability with the number of devices linked to a remote PC. Our work demonstrates an application-level solution for Bluetooth-based device synchronization without an undue burden on their hardware. With the development of the presented UI and the given NN-based solution, we demonstrate the possibility for unsynchronized wearable devices to operate in unison to enable multiple-limb, multi-person event-oriented motion capturing. The network can output high-frequency motion profiles captured at 200 Hz from multiple limbs without requiring sophisticated laboratory equipment. Thanks to the reported technology, in perspective, we could even connect heterogeneous wearable nodes using only standard Bluetooth technology through limited hardware additions (that is, the possibility of extracting the system clock tick) and a full application-level synchronization, thus easily providing improved sensor fusion (i.e., with electroencephalogram or electrocardiogram Bluetooth-based sensors) for the benefit of the scientific community.

## Methods
The research was approved by the local ethical committee (Comitato Etico Regionale della Liguria, under application `Prot. IIT_ERC_I-MOVEU_01`) and was carried out in accordance with the principles of the revised Helsinki Declaration[23].

### Motion capturing experiment
To effectively perform motion capture with a participant, the experiment has been divided into three steps, which are as follows: (1) Pre-experiment phase—The participant is thoroughly informed about the task that needs to be performed. Before attaching KiD to the participants' limbs, the device must be fully charged, the internal storage should be cleared, and it must be linked to the UI on the remote system. The neural network inside the UI will train the KiDs automatically and keep them ready for acquisition (see Supplementary Methods, "KiD Software flow chart"). (2) Motion capturing phase—An evaluator can use the UI to control the data collection. The `START` command initiates capturing, and the `STOP` command terminates the acquisition. The UI can start and stop the experiment and transmit the markers to identify tasks using the `MARK` command for extracting necessary artifacts. (3) Post-analysis phase—Finally, following data acquisition, the KiD is removed and wired to the remote system via micro USB for downloading the motion profile. Once the data has been transferred to the system, it can be associated with a machine learning technique or mathematical models to derive neuroscience outputs.

### Low-level parameter evaluation
Device sampling frequency ($f$) is a crucial low-level parameter to indicate the data/samples a sensor could acquire in one second. Not all devices operate at the same frequency due to several factors, such as the aging of the internal electronic components affected by temperature, power dissipation, and crystal drifts. The true value of the sampling frequency can be found while collecting samples for a specific time. By using the total of sample data from the device (i.e., the count of stored data $s_i$ in the device memory captured within the `START` and `STOP` commands, where $i$ is 1...3 represents a device number identifier) and capture event time ($T_i$), we can calculate the device sampling frequency $f_i = s_i/T_i$. The mean value ($\mu$) and the standard deviation ($\sigma$) for device sampling frequency can be obtained over an iteration of experiments.

### Evaluating wireless communication channel
During motion capture, KiDs were worn on a human limb and layered vertically, one on the other, to create the hand sequence (a hand motion that has to be captured), ensuring they had similar motion data to depict synchronization better. To evaluate the frequency drift among KiDs, we wrote a function that periodically reads the time from the devices and the remote system to obtain synchronization errors. For instance, we can read the time from the connected devices every 60,000 ms for an experiment. The function was repeated over time for several iterations to obtain statistical results, and this automated task was performed in the laboratory at a room temperature of 20–25 °C.

Bluetooth communication is a two-way transmission, i.e., any sent message always has an acknowledgment. For instance, let the remote system and device that communicate with each other using BLE transmit data. Two distinct events (one on the remote system and the other on the device) can occur, (1) two time-stamps on the remote system while sending $T^A$ and receiving $T^D$ a message at the system local time (RTC of the system) and (2) another two time-stamps at the device while receiving $T^B$ and acknowledging $T^C$ the message at device local time (RTC of the device). Time stamps indicate the occurrence time of events among the devices and the system. A synchronization event is the sequential transmission of multiple messages between wearables and a remote system to perform a synchronous task or activity. For illustration, let us consider a synchronization event having $i$ messages, their acquired time stamps can be written as: $\{T_i^A, T_i^B, T_i^C, T_i^D\}$. Let us consider transmission and reception as a time-bound operation. Hence, the time of execution at master is given as $r_A(t)$ and slave as $r_B(t)$, computed using Eq. (1),

$$t_i^{AD} = \frac{T_i^A + T_i^D}{2} ; t_i^{BC} = \frac{T_i^B + T_i^C}{2}, \tag{1}$$

$$r_{AD}(t) = t_i^{AD}, t_{i-1}^{AD}, \cdots t_2^{AD}, t_1^{AD}; r_{BC}(t) = t_i^{BC}, t_{i-1}^{BC}, \cdots t_2^{BC}, t_1^{BC}.$$

Actually, a synchronized message timing follows the relationship $r_{AD}(t) - r_{BC}(t) \approx 0$; that is, the clock frequency $\alpha$ and device offset $\beta$ remains constant ($\alpha \approx 1, \beta \approx 0$). Compactly, this can be expressed as shown in the following Eq. (2),

$$r_{AD}(t) = \alpha r_{BC}(t) + \beta. \tag{2}$$

In the real world, several non-idealities determine these hardware parameters $\alpha$ and $\beta$, thus impacting synchronization, as discussed previously,

$$r_{AD'}(t) = \alpha r_{BC}(t) + \beta. \tag{3}$$

CSPs always linearize the relationship between a generic master $M$ and a slave $S$, in particular, $r_M(t)$ and $r_S(t)$, thus allowing fine-tuning of the device operational parameters (device clock) for synchronization.

However, tuning operational parameters in a device in real-time would eventually lead to task error or unprecedented timing error that affects functionality, as many crucial functions rely on the device clock. Therefore, the synchronization error ($\tau$ from Eq. (3) and (4)) can be computed by finding the difference between time stamp $[r_M(t)]$ and the estimated value of it $[r_M^*(t)]$ and suitably compensate for this on the next message, in our case,

$$\tau = r_{AD}(t) - r_{BC}^*(t). \tag{4}$$

However, BLE is packet-based and non-continuous and never adopts a similar topology as that of wireless networks; that is, wireless networks have deterministic delays, which can be linearized. Specific properties of BLE that require the re-transmission of lost packets affect device latency. Therefore, the time difference between the connection interval and device latency remains a non-deterministic jitter ($\gamma$ see Eq. (5)). Considering the previous constraint, Eq. (2) can be rewritten as,

$$r_{AD}(t) = \alpha r_{BC}(t) + \beta + \gamma. \tag{5}$$

To observe this time-shift pattern and estimate the non-linear parameter above, we use a non-linear estimator that coherently recognizes any underlying relationship between two or more nodes. Here, the neural network is a three-layer perceptron with 20 Neurons, trained beforehand to the experiment with each time stamp of $r_{AD}$ and $r_{BC}$. From the time-stamp series, the neural network can thoroughly assess any non-idealities from both BLE and device that cause device offset,

$$r_{AD}^*(t) = v(t)r_{BC}(t). \tag{6}$$

Here, the neural network layer (see Supplementary Methods, "Neural Network concepts") operates as a virtual clock $v(t)$, as shown in Eq. (6)[24], which is a software clock for each node to implement device time (see Supplementary Methods, "Implementation technique"). The virtual clock layer computes the time difference between nodes to understand their underlying relationship and suitably compensates for the delay using Eq. (4). Here, the primary goal of the algorithm is to have a synchronization error lower than the data capture interval.

The training data sets are acquired from the training experiment when the devices are operating without any delays and adjusted at various situations within a 50 m range automatically by the UI. The root mean square error (RMSE) function is used in Python to train the model. RMSE calculates the input with the target values obtained from the KiD at various intervals using the TIME command. The error factor gradually reduces upon every epoch until it is minimized, thus reaching a suitable value that is good for synchronization. The three-layer neural network is designed to feature 20 hidden neurons. Further, this trained virtual clock layer is validated using the test data to verify appropriate operation before experimentation.

### Reporting summary
Further information on research design is available in the Nature Portfolio Reporting Summary linked to this article.

## Data availability
All data generated in this study are included in the manuscript and the Supplementary Material. Source data are provided with this paper.

## Code availability
Supplementary Code files show how to interface the wearable devices, carry out experiments, and gather data from the local system. Additionally, the supplementary material offers comprehensive illustrations of the methods to implement it on a wearable device.

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

## Acknowledgements

The European Research Council supports this research under the ERC-2017-PoC grant agreement 789601 to C.B., and N.C.F. received funding from the European Union's Horizon 2020 research and innovation program under the Marie Sklodowska-Curie grant agreement No. 754490-MINDED project. Participants provided written consent before recording their data. We extend our gratitude to those who volunteered for data recording. Also, we thank Davide Dellepiane, IIT-Electronic Design Laboratory, for the assembly and mechanical/soldering works.

## Author contributions

C.B. conceived and supervised the study. C.B., M.C., and A.C. conceptualized the wearable movement measurement platform. M.C. supervised the design and manufacturing of the device. G.Z. and A.M. designed the device. K.K.B. and A.M. engineered the firmware and implemented the software platform with a dedicated graphical user interface under the supervision of M.C. and with inputs from N.C.F. and A.C., then K.K.B. analyzed the data with inputs from M.C., A.M., G.Z., A.C., N.C.F., C.B., and K.K.B. and M.C. wrote the manuscript.

## Competing interests

The authors declare no competing interests.
