## [Peer Review File · Nature Communications]

Neural network-based Bluetooth synchronization of multiple wearable devicesREVIEWER COMMENTS

Reviewer #1 (Remarks to the Author):

This paper reports a synchronous network that allows many Bluetooth based wearable to connect and work together to obtain task based motion data. There are several important problems in the article that need to be solved by the author.

1. This paper proposes a network that allows Bluetooth based wearable to work synchronously, but there is no specific experimental results to quantify the average synchronization error after optimization. In the following paper, the author shows that the average synchronization error as low as 320 ns/60 s (single hop) can be achieved through experiments.

IEEE INTERNET OF THINGS JOURNAL, VOL. 9, NO. 11, JUNE 1, 2022

2. This study adds a new function called "label" to KiD, which provides a method for classifying motion data. Can you add more detailed text to explain how "label" benefits from higher level motion interpretation, classification and clustering. Reviewers personally understand that the role of "label" is to facilitate data segmentation.

3. Low power consumption is an important part of the Bluetooth protocol. Please add whether the neural network at the interface increases the low power consumption Bluetooth communication overhead between remote systems and devices

4. In this paper, a three-layer perceptron with 20 neurons is used to fine tune the virtual clocks of multiple wearable devices to achieve mutual synchronization. The neural network acts as the virtual clock. As the innovation of this paper, the author ignores the detailed simulation results of the neural network, and should put it in a more important position to describe, such as the graph of the correlation between the actual value and the predicted value changes with the number of iterations, the graph of the change of the mean square error, etc.

5. The main content of this paper is to achieve synchronization of multiple wearable devices. Whether there is some exaggeration in the topic of this paper for neuroscience study, the experimental simulation in this paper has no direct relationship with neuroscience.

6. This paper reports that an intuitive interface provides mutual synchronization of wearable devices, and whether other types of commercially available chips can be used to verify the universality of the method proposed in this paper.

Reviewer #2 (Remarks to the Author):

The paper presents design methodology and functionality of a wearable device (Kinematics Detector-KiD) that might be used in the evaluation of sensorimotor behaviour of objects/human, or, perhaps simply in any human-comparable motion capture application. The major components of the wearable are described regarding with high level technical requirements and constraints, and also various technology alternatives are assessed comparatively. Specifically, technical limitations toward synchronisation is discussed.

As far as I see, the technology adapted is very-well known and mature, and all are widely used in various similar devices (both HW and SW tools, components/modules are cheap COTS). However, I have the following comments:

-When the comparison table (Fig.8) is inspected, it seems that the device is mostly superior to all other competing devices in the market. The major issue, here, would be why battery time could be low. Is this due to the size/weight trade off, or technology used (BLE or ucontroller or any continuously operating mode or functions, or what?)? It could be justified.

-The paper discussed technical challenges and alternatives for synchronization. However, it simply say, at the end, that neural network is preferred but not much is given about how it is implemented, or how it provides an "acceptable outcome". The details of implementation would be discussed.

-In my opinion, prices (or somehow "relative" numbers if not possible) might also be a parameter if such detailed list (fig.8) is given. Note that the table states "devices in the market". Any device might be superior but much expensive than others (very critical decision parameter).

Reviewer #3 (Remarks to the Author):

This paper presents the use of a neural network to synchronize data from bluetooth data sensors.

The work appears to be sound and noteworthy, however in it's current form it is quite difficult to follow. Consider the section headings alone, there is no "Introduction" or "Background", but rather a "Main" that appears to be a subheading of the abstract based on font sizes? Then the "Conclusion" arrives before any discussion of the actual methods used to perform the synchronization, leading to reader to think they have missed it and re-reading before eventually discovering the actual synchronization method appears at the very end. This unusual structuring also leads to some unnecessary repetition of information.

I think the work is publishable, but is in need of some restructuring and editing for readability.

Note for reviewers

We thank all the reviewers for the time dedicated in reviewing our manuscript.

Additionally, we have provided a revised manuscript (NN_revision.pdf) with inline modifications in red and removed text parts canceled (for example ~~this~~) and embedded with indications of line numbers. The line numbers below refer to this manuscript version with indicated modifications. The said red highlighting does not apply to the references.

As requested by the journal rules, we have provided the revised manuscript (NN_revision_clean.pdf).

Please find below each point-to-point response.

Reviewer # 1

Reviewer # 1, Comment: This paper reports a synchronous network that allows many Bluetooth based wearable to connect and work together to obtain task based motion data. There are several important problems in the article that need to be solved by the author.

Author response: Thank you for reading our manuscript and dedicating your precious time for the review. Our primary objective is to connect numerous Bluetooth-based edge wearables to work together for capturing physiological data. Here, to explain the synchronization mechanism, we use Kinematics Detector (KiD, an edge device for capturing toddlers' limb movements), which has Bluetooth Low Energy 4.2 module and internal storage memory. We have also designed a new User Interface (UI) for such devices, which introduces a concept called application-based Bluetooth synchronization. We have considered every valuable point of the reviewer and modified the manuscript accordingly to provide better readability.

Reviewer # 1, Concern 1: This paper proposes a network that allows Bluetooth based wearable to work synchronously, but there is no specific experimental results to quantify the average synchronization error after optimization. In the following paper, the author shows that the average synchronization error as low as 320 ns/60 s (single hop) can be achieved through experiments.

IEEE INTERNET OF THINGS JOURNAL, VOL. 9, NO. 11, JUNE 1, 2022.

Author response: Thank you for highlighting this perspective. We have now added a method to quantify the synchronization error and reported the average synchronization error in the manuscript. In the aforementioned paper, the authors have used Generic Access Profile (GAP) or advertisers from the BLE specifications to achieve synchronization. Moreover, the authors suggest adjusting hardware time parameters during the device's runtime could improve Bluetooth synchronization, where this work is admirable.

We want to inform the reviewer that we use Generic Attribute (GATT), a different approach than GAP. In contrast to GAP, GATT typically uses an entirely different methodology that relies on a one-to-one network transmission. Owing to data security, we chose this method, offering custom functions and properties to read and write data wirelessly, and we then explained how to utilize them in the manuscript. Also, adjusting the device's low-level time parameters may eventually make them unreliable. Since there is no mature solution for device synchronizing, we present an application-based Bluetooth synchronization.

Author action: We have added the following lines in the manuscript that discusses the methods to quantify the average synchronization error from LINES: 248 to 252. Moreover, We have now added

an elaboration on the BLE connective modes, that is, GAP and GATT from the BLE protocol in the LINES: 47 to 60.

Reviewer # 1, Concern 2: This study adds a new function called "label" to KiD, which provides a method for classifying motion data. Can you add more detailed text to explain how "label" benefits from higher-level motion interpretation, classification and clustering. Reviewers personally understand that the role of "label" is to facilitate data segmentation.

Author response: Thank you for having pointed this out. We have now emphasized the event or artifact extraction that addresses the need for a label, and how to implement and collect the data using the labels.

Author action: We have revised the subsection and renamed it to "Artifact extraction and labeling" in the manuscript at Lines 271 to 291.

Reviewer # 1, Concern 3: Low power consumption is an important part of the Bluetooth protocol. Please add whether the neural network at the interface increases the low power consumption Bluetooth communication overhead between remote systems and devices.

Author response: We thank the reviewer for pointing out the right method to describe the importance of low power consumption using the application-based synchronization method.

Author action: We have now included in the "discussions" what is the impact of Neural Network on power consumption while synchronizing wearables at the LINES: 309 to 348.

Reviewer # 1, Concern 4: In this paper, a three-layer perceptron with 20 neurons is used to fine tune the virtual clocks of multiple wearable devices to achieve mutual synchronization. The neural network acts as the virtual clock. As the innovation of this paper, the author ignores the detailed simulation results of the neural network, and should put it in a more important position to describe, such as the graph of the correlation between the actual value and the predicted value changes with the number of iterations, the graph of the change of the mean square error, etc.

Author response: We thank the reviewer for highlighting this point.

Author action: We have now added the neural network simulation results onto the supplementary material under the subsection "Neural Network concepts."

Reviewer # 1, Concern 5: The main content of this paper is to achieve synchronization of multiple wearable devices. Whether there is some exaggeration in the topic of this paper for neuroscience study, the experimental simulation in this paper has no direct relationship with neuroscience.

Author response: We thank the reviewer for pointing out that the central message of the paper is more of a neuroscience.

We designed a tool for controlling the motion capture not bound to the laboratory but in a real-world scenario. Here, data capturing will have massive volumes of data (accelerometer profiles); if it is not categorized at the source, it would be difficult to interpret a specific task afterward. Thus, we have added a new feature to the wearable that allows sending labels to the source directly while capturing. These labels can be sent to the device or the synchronized network with multiple devices for event-based motion capture. The acquired data is ready for analysis, significantly reducing time while post-processing.

Author action: We have now modified the manuscript or toned down the neuroscience part of the manuscript to focus more on our innovation.

Reviewer # 1, Concern 6: This paper reports that an intuitive interface provides mutual synchronization of wearable devices, and whether other types of commercially available chips can be used to verify the universality of the method proposed in this paper.

Author response: We presume the reviewer intends to use commercial products to ensure universality. On the other hand, commercial products prevent end-users from altering or customizing their low-level implementations. However, we have illustrated and demonstrated the technology behind synchronization in this manuscript. Moreover, the supplementary article contains design concepts, implementation approaches, a software flowchart, and configuration instructions. With these methods and implementations, this application-based synchronization strategy can be extended to other wearables of homogeneous or heterogeneous sensor types (for instance, EEG and IMU sensors), opening up a new possibility for capturing relative data.

Author action: We have now precisely added the methods and strategies to the supplementary material of the manuscript that provides more on software methodology and hardware re-configurations needed for synchronization.

Reviewer # 2

Reviewer # 2, Comment: The paper presents design methodology and functionality of a wearable device (Kinematics Detector-KiD) that might be used in the evaluation of sensorimotor behaviour of objects/human, or, perhaps simply in any human-comparable motion capture application. The major components of the wearable are described regarding with high level technical requirements and constrictions, and also various technology alternatives are assessed comparatively. Specifically, technical limitations toward synchronisation is discussed.

As far as I see, the technology adapted is very-well known and mature, and all are widely used in various similar devices (both HW and SW tools, components/modules are cheap COTS).

Author response: We thank the reviewer for his considerable time in reviewing our manuscript. As the reviewer has rightly pointed out, the manuscript describes the methodology, functionality, and testing of application-based synchronization for Bluetooth-enabled wearables. Nevertheless, we would like to highlight the novelty of our process once again. We proposed an application-based synchronization method in the remote system without any requirements for configuring or altering any low-level time parameters in the hardware for synchronization. It can, however, be utilized in any wireless network using the current format. Thus far, conventional algorithms for synchronization have been demonstrated while modifying the low-level wearable time parameters, which could preserve time consistency among numerous devices. Wearable programming in this mode requires intricate technical planning, involves additional hardware parameters, and is best accomplished through bare-metal programming.

Bare-metal programming has been the best method for decades, providing extreme customization and control over the microcontrollers, notwithstanding a high design time and code cost. Adopting Industrial standardization, reliability, and supportability always remain a question. For instance, a small parameter change in the software or hardware demands an additional layer for testing. On the other hand, RTOSs continue to enjoy widespread industry acceptance for their simple system implementation, deterministic performance, reliable and secure task management, and scalability.

While stressing a few benefits of RTOS, the low-level time parameter alteration at RTOSs would compromise the device reliability, particularly in (i) Timer-critical subroutines: Waiting for a certain amount of time before performing a task or triggering an event would fail concerning the device change that could affect operation timing, causing them to happen earlier or later than expected; (ii) Event synchronization: Events and semaphores to synchronize tasks and communicate between them fail because device time change may affect them, causing tasks to wait longer or shorter than expected; (iii) Multi-task scheduling: Multi-tasks work based on time-slicing concepts when the device time or

clock is altered, which would eventually miss critical interrupts or send untimely signals; (iv) Time-stamped data: In some devices, data is time-stamped to record when it was created or updated. Changing the device time may affect the accuracy of these time stamps, causing data to appear out of order or with incorrect time stamps.

Considering all these facts, i.e., the limitations in the conventional algorithms for synchronization, tedious bare-metal programming making the development chain below a for close examination for correctness, and while accepting the difficulties of the RTOSs, we demonstrate Bluetooth synchronization on multiple wearables that specifically run based on RTOSs at the wearables, while overcoming all the shortcomings using application-based synchronization.

Reviewer # 2, Concern 1: When the comparison table (Fig.8) is inspected, it seems that the device is mostly superior to all other competing devices in the market. The major issue, here, would be why battery time could be low. Is this due to the size/weight trade off, or technology used (BLE or ucontroller or any continuously operating mode or functions, or what?)? It could be justified.

Author response: We thank the reviewer for pointing this out. The primary focus of this paper was Bluetooth synchronization for wearables. For a demonstration of synchronization, we use KiD, which has a 100mAh battery for a simple neuroscience experiment for toddlers. KiD was conceived in this aspect "for toddlers", so it had to have a perfect trade-off between weight, size, and operation time. The overall weight of KiD is 10 g without the strap, and size (35×20×10 mm³). We have sized the battery to accommodate 150 min of continuous experimentation.

Author action: We have added a subsequent paragraph to the supplementary material addressing the size/weight trade-offs and the design protocols while designing wearables in the section "Wearable device safety measures for toddlers."

Reviewer # 2, Concern 2: The paper discussed technical challenges and alternatives for synchronization. However, it simply say, at the end, that neural network is preferred but not much is given about how it is implemented, or how it provides an "acceptable outcome". The details of implementation would be discussed.

Author response: We thank the reviewer for emphasizing this point.

Author action: We have now revised the manuscript to focus more on how it is implemented and how to expect an acceptable outcome from LINES: 175 to 267

Reviewer # 2, Concern 3: In my opinion, prices (or somehow "relative" numbers if not possible) might also be a parameter if such detailed list (fig.8) is given. Note that the table states "devices in the market". Any device might be superior but much expensive than others (very critical decision parameter).

Parameters	WITMOTION	3 Space Bluetooth	Perception Neuron Pro	XSENSE DOT	This Work KID
Price/Cost per device (euros)	N.A.	€237	€400	€150	€137*#
Relative pricing/cost (times)	N.A.	1.75	2.91	1.094	1*#
* The cost listed is for the prototypes used in this work. However cost can further decrease with larger production volumes.					
# Including the complete electronics fabrication without mechanical parts.					

Author response: Thank you for pointing us in comparing the price parameter among devices, which is obviously a critical parameter as said. We would like to let the reviewer know that KiD is yet a prototype, the production costs projected here include the cost of manufacturing a few devices.

Nonetheless, the cost of manufacturing would be considerably reduced while producing large volumes.

Author action: We have now updated the comparison table to reflect the new specifications from the online-available datasheets and the price parameters have also been added to Figure 8.

Reviewer # 3

Reviewer # 3, Comment: This paper presents the use of a neural network to synchronize data from bluetooth data sensors.

Reviewer # 3, Concern 1: The work appears to be sound and noteworthy, however in it's current form it is quite difficult to follow. Consider the section headings alone, there is no "Introduction" or "Background", but rather a "Main" that appears to be a subheading of the abstract based on font sizes? Then the "Conclusion" arrives before any discussion of the actual methods used to perform the synchronization, leading to reader to think they have missed it and re-reading before eventually discovering the actual synchronization method appears at the very end. This unusual structuring also leads to some unnecessary repetition of information.

I think the work is publishable, but is in need of some restructuring and editing for readability.

Author response: Thank you for reading our manuscript and giving us your valuable remarks. We are delighted to hear that you found our manuscript interesting. We have now revised the manuscript as per your recommendation.

Author action: We have now included the subsection "Discussion." Also, we have restructured and edited the manuscript for better readability.

REVIEWERS' COMMENTS

Reviewer #1 (Remarks to the Author):

This paper proposed a synchronization network in which many Bluetooth-based wearables can be connected and work together to obtain event-oriented sensor data.

I value the authors efforts to answer all the previous concerns and feel that the paper overall improved. It seems to me that most of the previous concerns were well addressed in the revised manuscript. Overall, the revised paper quality largely meets the requirements of "Nature Communications", hence the manuscript could be accepted for publication after minor polish.

Reviewer #2 (Remarks to the Author):

I examined the relevant sections of the revised paper, and then overall. As far as I see, the authors have addressed two of my major concerns, #2 to #3, the details of NN based synch and price issues.

On the other hand, I noticed that eqn 1 to 4 seem to be weird for scientific papers, in the revised paper. It is not a correct convention to use symbol with the parameter name in an equation (first define symbol, then use it in an equation). It should be corrected in my opinion.

The paper can be published then.

Note for reviewers

We thank the reviewers for their timely review of our manuscript.

Additionally, we have provided a revised manuscript (NN_revision.pdf) with inline modifications in red and removed text parts canceled (for example ~~this~~) and embedded with indications of line numbers. The line numbers below refer to this manuscript version with indicated modifications. The said red highlighting does not apply to the references.

As requested by the journal rules, we have provided the revised manuscript (NN_revision_clean.pdf).

Please find below each point-to-point response.

Reviewer # 1

Reviewer # 1, Comment: This paper proposed a synchronization network in which many Bluetooth-based wearables can be connected and work together to obtain event-oriented sensor data. I value the authors efforts to answer all the previous concerns and feel that the paper overall improved. It seems to me that most of the previous concerns were well addressed in the revised manuscript. Overall, the revised paper quality largely meets the requirements of "Nature Communications", hence the manuscript could be accepted for publication after minor polish.

Author response: Thanks a lot to the reviewer for reviewing the manuscript. The suggestions and comments from the reviewer have allowed us to improve the manuscript's central idea.

Author action: We have now revised the manuscript with a few minor corrections as shown in the NN_Revision.pdf.

Reviewer # 2

Reviewer # 2, Comment: I examined the relevant sections of the revised paper, and then overall. As far as I see, the authors have addressed two of my major concerns, #2 to #3, the details of NN based synch and price issues.

Author response: We thank the reviewer for giving us recommendations, important points, and fine particulars that enhanced the manuscript's clarity. We would like to inform the reviewer that Kinematics Detectors are primarily designed for toddlers. Considering the safety factors while using the device for neurological assessments, we required a lightweight device. Thus KiD has a minimal size and weight with a limited battery size (i.e., a shorter time was sufficient for a simple neurological experiment).

Author action: We have added the lower battery time justification both in the Supplementary Information and the main manuscript.

1. In **Supplementary Information:** We have added an intercept to the Figure 8 discussing parameter comparison.
Size and weight constraints limit the battery capacity. The battery time of KiD is still enough for seamless operation during a typical recording session.
2. In **Supplementary Information:** We have added an additional paragraph on **Safety measures: wearables for toddlers.**

Safety factor recommendations while designing wearable devices for toddlers are a crucial design criterion, (i) Neutral color: Children beyond four months naturally tend to attract colors. Hence, a neutral color must be chosen to discourage the child’s attention. (ii) Size: Wearables should be conventionally larger than the children’s typical esophagus³⁸. Moreover, there should be a safety hole for allowing respiration in case of an accident. (iii) Accessories: Wearables should not have removable accessories (e.g., access media or battery). (iv) Material: Wearables must be made of materials that are devoid of allergens, strong, and resistant to wear and tear. Kinematics Detector (KiD), by default, has two-layer protection, a silicone outer cover has a particular color (i.e., KiD is grey colored following a neutral color policy) to discourage the child’s attention and a Plastic inner case to limit the circuitry access to the child², as shown in Fig. 7a. This protective outer layer is meant to be larger than a children’s typical esophagus³⁸ to avoid accidental swallowing. Moreover, KiDs do not need external media storage, such as a Secure Digital card for storing motion profiles, to avoid removing the media after every usage. Instead, we have designed KiDs with onboard memory to save motion profiles. KiD provides straightforward wearability (volume of $35 \times 20 \times 10 \text{ mm}^3$, and weight 10 g) so that the devices can be worn on the toddlers’ limbs (e.g., wrist, arm, leg, torso, and ankle, see the identified at positions 1, 2, 3, and 4 in Fig. 7b) for capturing the associated motion profile. Considering safety factors, short-lived experiments are advisable for toddlers, and hence a battery capacity of 100 mAh can be enough for a simple neurological experiment among toddlers.

3. We have added the following intercept to the LINES: 257 to 264 of the main manuscript.

Our work demonstrates the capability of NN layer to solve synchronization issues at the application level. The proposed solution requires additional Bluetooth-transferred data to acquire clock ticks for time stamp recording during a command transfer. In this context, a wearable must draw additional power for every extra communication the remote system requires. However, our data show that the additional Bluetooth communication load with each wearable KiD device was limited and did not impact battery life, which was unvaried compared to the initial design requirements, i.e., 150 min acquisition session.

Reviewer # 2, Concern 1: On the other hand, I noticed that eqn 1 to 4 seem to be weird for scientific papers, in the revised paper. It is not a correct convention to use symbol with the parameter name in an equation (first define symbol, the use it in an equation). It should be corrected in my opinion.

Author response: Thanks for the reviewer emphasizing on the equations. We have now revised the paragraph and removed the redundant equations.

Author action: We have revised the paragraph **Low-level parameter evaluation** from LINE: 292 in the main manuscript.

Device sampling frequency (f) is a crucial low-level parameter to indicate the data/samples a sensor could acquire in one second. Not all devices operate at the same frequency due to several factors, such as the aging of the internal electronic components affected by temperature, power dissipation, and crystal drifts. The true value of the sampling frequency can be found while collecting samples for a specific time. By using the total of sample data from the device (i.e., the count of stored data s_i in the device memory captured within the START and STOP commands, where i is 1...3 represents a device number identifier) and capture event time (T_i), we can calculate the device sampling frequency $f_i = s_i/T_i$. The mean value (μ) and the standard deviation (σ) for device sampling frequency can be obtained over an iteration of experiments.